# Who Pays the Bill? Assessing Ecosystem Services Losses in an Urban Planning Context

Harald Zepp [1,*] and Luis Inostroza [1,2] 

1   Institute of Geography, The Ruhr-University Bochum, 44801 Bochum, Germany; luis.inostroza@ruhr-uni-bochum.de
2   Universidad Autonoma de Chile, 4810101 Temuco, Chile
*   Correspondence: Harald.Zepp@ruhr-uni-bochum.de

**Abstract:** While Ecosystem Services (ES) are crucial for sustaining human wellbeing, urban development can threaten their sustainable supply. Following recent EU directives, many countries in Europe are implementing laws and regulations to protect and improve ES at local and regional levels. However, urban planning regulations already consider mandatory compensation for the loss of nature, and this compensation is often restricted to replacing green with green in other locations. This situation might lead to the loss of ES in areas subject to urban development, a loss that would eventually be replaced elsewhere. Therefore, ES assessments should be included in urban planning to improve the environmental conditions of urban landscapes where development takes place. Using an actual planning and development example that involves a proposed road to a restructured former industrial area in Bochum, Germany, we developed an ad-hoc assessment to compare a standard environmental compensation approach applying ES. We evaluated the impact of the planned construction alternatives with both approaches. In a second step, we selected the alternative with a lower impact and estimated the ES losses from the compensation measures. Our findings show that an ES assessment provides a solid basis for the selection of development alternatives, the identification of compensation areas, and the estimation of compensation amounts, with the benefit of improving the environmental quality of the affected areas. Our method was effective in strengthening urban planning, using ES science in the assessment and evaluation of urban development alternatives.

**Keywords:** compensation measures; urban resilience; urban development; impact assessment

## 1. Ecosystem Services for Cities

Ecosystem Services (ES) science has provided a framework and empirical evidence for analysing, discussing and communicating environmental trade-offs arising from alternative development options in several planning contexts [1–3]. Resilience, sustainability and quality of life can be greatly improved in urban areas by including ES assessments [4,5]. Urban planning can benefit from the adequate use of the ES framework. Urban areas need to improve the application of laws and regulations following EU-directives to protect and improve the natural environment at local and regional levels [6,7].

This paper uses the ES concept in a practical way to illustrate its potential to support decision-making. The aim is to present a clear way to apply the ES framework in an actual urban development case to show the benefits of such an approach. We apply the ES concept in a real planning case study using a procedure that is ready to implement and easy to understand, avoiding unnecessary conceptual and operational complexity. Our approach can be used by a broader community of scholars and decision-makers who might not necessarily be familiar with the ES concept. We present the ES concept, focusing on how it can be implemented in practice for urban planning and the necessary steps to follow.

## 1.1. Ecosystem Services Assessments

ES are the benefits that society obtains due to the functioning of healthy ecosystems [8]. ES can be classified to assist in assessing them. The Economics of Ecosystems and Biodiversity (TEEB) and the Common International Classification of Ecosystem Services (CICES) [9] are two broadly accepted classification systems. We used the CICES, which classifies ES into three groups: provisioning Ecosystem Services (P-ES), regulating Ecosystem Services (R-ES), and cultural Ecosystem Services (C-ES) [10,11]. CICES version v5.1 offers a detailed and extensive list of ES that can be applied to identify relevant services in several geographical settings.

In the context of urban planning, the assessment of ES should always start with screening, identifiing and selecting relevant ES. This is a fundamental starting point, because ES are always context-specific. This means that the presence, intensity, distribution, and relevance of ES change from location to location [12]. ES also change due to land management practices or urbanisation intensity [12,13]. A good practice for sound identification is to refer to an established classification system, such as TEEB or CICES. Using a validated classification of ES can help ensure a systematic screening process that does not leave out any important service, and that help avoid the inclusion of benefits that are not ES. Having a consolidated list of relevant ES a follow-up good practice is to perform an exploratory assessment to provide a sound evaluation of ES intensities that can help in further prioritizing and mapping ES. Many studies have applied these two steps using expert assessments and the matrix approach [3,14,15]. A pool of experts can select relevant ES in a study area and score the intensity of the ES. The matrix approach can link such scores to specific land use/land cover (LULC) to map the spatial distribution of ES. Preliminary scoring and mapping of ES using the matrix approach have been shown to have high concordance with biophysical estimations [16]. These steps can be a robust guide in further evaluations and biophysical quantifications of ES in a more detailed analysis.

## 1.2. Ecosystem Services for Spatial Planning

The capacity of the ES framework to assist in decision-making in several fundamental aspects of urban development, such as green infrastructure, climate change adaptation and sustainable urban development, has been largely confirmed [4,17–20]. ES can greatly help planners understand the dynamics of decision-making in complex eco-sociotechnical systems [21]. In concrete terms, in the context of planning, ES knowledge can have both conceptual and instrumental uses. The conceptual use of ES is aimed at broadening understanding to shape decision-makers' and stakeholders' thinking. The instrumental use of ES is focused on supporting the decisions between policy options regarding gains and losses, and involving concrete decision options [22]. To have practical instrumental value, ES information must be presented in a meaningful manner [23] and be ready to be applied in real-world situations. However, there is a need for feasible methods, models and applications that can assist planners in the practical implementation of ES science [21]. Empirical evidence shows that there are several problems, such as data availability, uncertainties, and, most importantly, difficulties in translating abstract scientific knowledge into practical applications and linking assessments to the characteristics of a specific local context, that planners face when attempting to use ES knowledge [24]. Furthermore, the ES concept cannot easily be translated into a legal framework and technical guidelines established as routine workflows in cities and regions [18].

The ES framework can link environmental aspects under an urban development perspective to better understand their effects on human wellbeing [25]. Integrating ES into urban planning can provide important benefits, such as (1) supporting the implementation and design of adequate measures to address current urban challenges, such as climate adaptation; (2) enhancing the transparency of trade-offs and cobenefits arising from urban development, while increasing awareness about the hidden or underrepresented values of nature that could eventually be lost; and (3) directly addressing issues of environ-

mental justice in land-use change decisions through the identification of ES demand and supply [19].

The need to incorporate ES into urban planning is not only related to the desire to improve practice with new knowledge. The inclusion of ES in policymaking has already resulted in important policy recommendations in the EU. The EU Biodiversity Strategy for 2030 explicitly addresses the need to incorporate ES mapping, monitoring and assessing into policy making; action 7 aims to ensure no net loss of biodiversity and ES [26]. The Territorial Agenda, a strategic policy document for guiding the spatial planning in regions and communities in Europe, explicitly highlights the relevance of ES to ensure their provision and public awareness of them [27]. Finally, the EU guidance on integrating ecosystems and their services into decision-making outlines concrete actions for the integration of ES into a range of decisions at different levels and areas, including spatial planning. This report emphasizes the inclusion of ES within existing planning frameworks to avoid the generation of parallel processes and assessments [28]. There is a set of criteria for addressing the potential negative impacts on ES. This mitigation hierarchy includes: *"(1) Avoidance: measures to identify and completely avoid detrimental impacts from the outset, such as careful spatial placement of infrastructure; (2) minimisation: measures to reduce the duration, intensity and/or extent of detrimental impacts (including direct, indirect and cumulative impacts) that cannot be completely avoided; (3) rehabilitation/restoration: measures to rehabilitate degraded ecosystems or restore cleared ecosystems following impacts that could not be completely avoided and/or minimised; (4) offsetting: measures to compensate for residual, significant, adverse impacts that could not be avoided, minimised or restored. Measures to overcompensate for losses can also lead to net societal gains by their contribution to well-being and prosperity"* [29]:13. This mitigation strategy is aimed at ensuring an increased delivery of multiple ES. On the other hand, according to this report, only a *"few cities have prioritised access to nature as a central objective of urban planning."* [28].

This paper addresses three research questions: (1) how can the ES framework be methodologically and operationally incorporated into urban planning? (2) How can the results of ES assessments be translated into urban planning tools for the public, stakeholders, and decision-makers? (3) Can ES help avoid the environmental deterioration occurring due to urban development?

## 2. Materials and Methods

The methods were designed for the instrumental use of ES, which supports decisions between policy options regarding gains and losses, and involves concrete decision options [22]. Our approach aimed to provide a sound assessment of environmental compensation accounting for the eventual loss of ES due to urban development. The approach was based on the analysis and selection of the best planning alternative by weighing the impacts of each development option in a comparative approach. The positive and negative environmental effects of the project to be implemented were investigated and compared to choose, modify, or reject the planning ideas. We used a double method to assess the impacts of the planning ideas. In the first assessment, we used a standard approach to calculate the environmental compensations; in the second assessment, we used the ES framework. We illustrated our method by using an example from Germany's Ruhr region. Our method is unique in that it articulates ES knowledge with a practical application based on a real planning situation, showing how the ES framework can support decision-making.

### 2.1. Case Study

The city of Bochum has 371,000 inhabitants, and it is part of Germany's Ruhr metropolis, which is one of the largest metropolitan areas in Europe with 5.1 million inhabitants. During the 19th and 20th-centuries, coal mining and steel production were fundamental economic activities. For the last 50 years, structural economic change has driven the closure of all coal mines and resulted in a decrease in steel production. This situation has transformed the economic base of the city to electronic devices manufacturing and

car production, with a more diversified sectoral mix, including service industries and universities, as a part of the knowledge-based economy.

The study area is located in the eastern part of Bochum. It can be considered a typical example of "glocalisation" [30], depicting the local effects of economic globalisation. The multinational GM/Opel car factory is located at two sites within the city of Bochum, covering an area of approximately 100 hectares. After the company decided to end car production at the end of 2015, one site was given up, and the other site was developed to serve as the European logistics centre for the distribution of spare car parts and a new industrial area. The existing access road connecting the site by the freeway crosses a residential area, and will become overloaded by increasing traffic. To ameliorate the environmental impacts, four alternative corridors have been discussed. The four planning alternatives for the access roads to the former factories are generally presented in Figure 1.

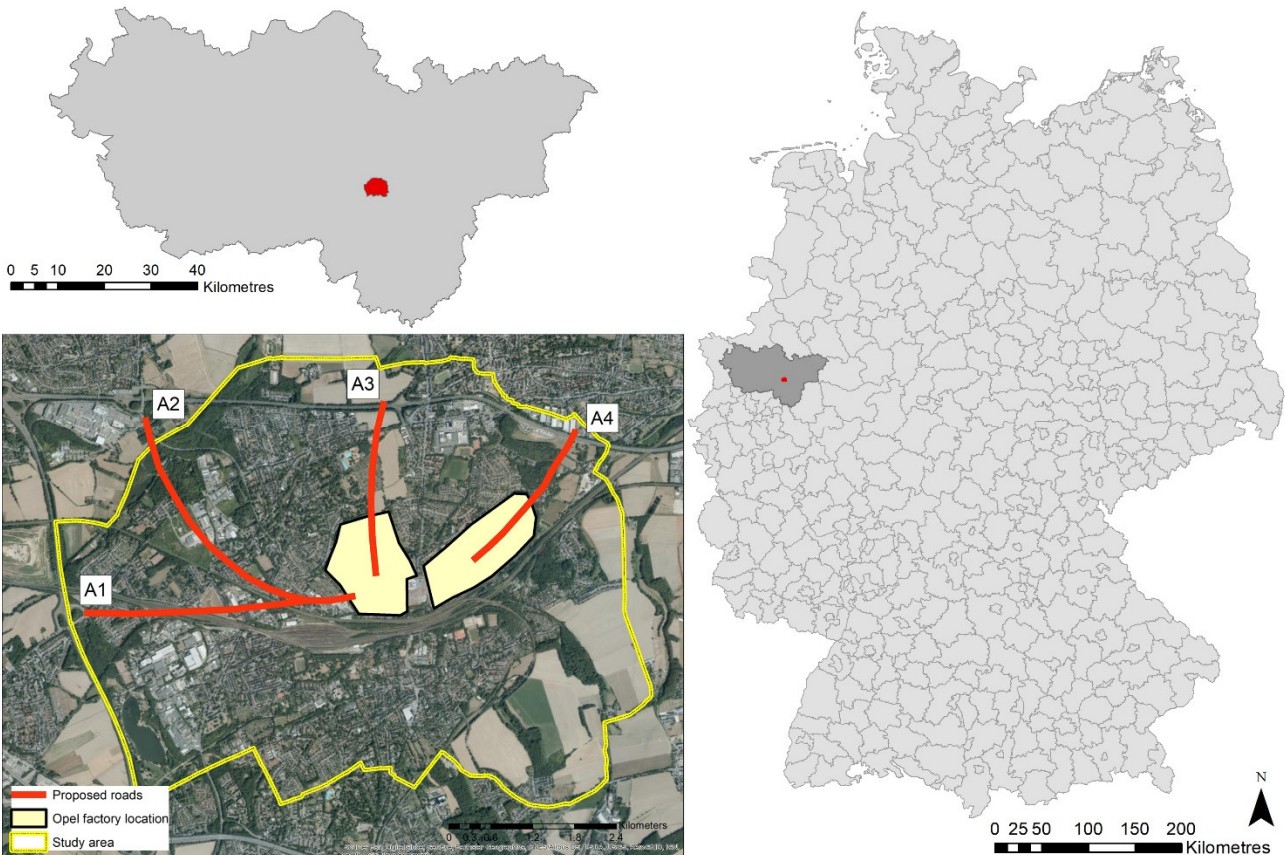

**Figure 1.** Location of the study area in Germany and the Ruhr area, and main features. Four alternative access roads to the GM/Opel European logistics centre are under discussion (A1, A2, A3 and A4). Satellite imagery: ESRI, DigitalGlobe, GeoEye, Earthstar Geographics, CNES/Airbus DS, USDA, USGS, AeroGRID, IGN, and the GIS User Community.

### 2.2. Mapping Urban Structural Types

We delineated the area named Lagendreer-Werne with a total surface area of 1632 hectares. We mapped the entire area in several field campaigns. This detailed mapping effort yielded several LULC classes. We grouped the LULC according to 22 urban structural subtypes (USSs), representing urban morphological units that embody the characteristics of the urban structure. The characteristics of urban structural types were related to factors such as the surface materials, the internal configuration of diverse open and sealed patches, the height of vegetation, and height [31]. We further differentiated the USSs listed in Table 1. A visual field survey was conducted after preparatory mapping based on aerial photos. Among the USSs, we identified open space, which normally contains the highest environmental values (Table 1).

**Table 1.** List of USSs and ES values used in the assessment. P-ES: provisioning ES; R-ES: regulating ES; C-ES: cultural ES. Column *p* indicates USSs that are terrestrial open space on natural soils. Values for P-ES, R-ES and C-ES are the mean of the single ES in each of the P-ES, R-ES and C-ES groups. The range is 0 to 5. High values indicate high ES supply provided by the USSs.

| *i* | Urban Structural Subtypes | P-ES | R-ES | C-ES | *p* |
|---|---|---|---|---|---|
| 1 | Allotment gardens | 2.1 | 3.4 | 3.6 | 1 |
| 2 | Arable field | 2.0 | 2.8 | 2.3 | 1 |
| 3 | Cemeteries | 1.1 | 3.3 | 3.5 | 1 |
| 4 | Commercial and industrial uses with a high degree of surface sealing | 0.8 | 1.0 | 0.7 | |
| 5 | Detached and semi-detached houses | 1.4 | 2.0 | 1.2 | |
| 6 | Housing complexes with green areas, e.g. row houses single multi-story houses | 1.2 | 1.6 | 0.8 | |
| 7 | Lakes ponds | 2.1 | 2.5 | 3.8 | |
| 8 | Linear groves (including industrial & green buffers of industrial areas) | 0.9 | 2.8 | 2.2 | |
| 9 | Mixed commercial and residential uses with a low degree of surface sealing | 0.9 | 1.3 | 0.8 | |
| 10 | Parking lots and parking areas | 0.2 | 0.4 | 0.3 | |
| 11 | Parks and green belts | 1.4 | 3.8 | 4.1 | 1 |
| 12 | Pasture and meadow | 2.0 | 3.7 | 3.6 | 1 |
| 13 | Places and squares | 0.3 | 0.4 | 0.5 | |
| 14 | Playgrounds | 0.6 | 1.5 | 2.3 | |
| 15 | Public buildings and public institutions | 1.0 | 1.4 | 1.5 | |
| 16 | Railroad tracks | 0.3 | 1.0 | 0.9 | |
| 17 | Roads | 0.2 | 0.4 | 0.3 | |
| 18 | Sports grounds and leisure infrastructure | 0.7 | 1.4 | 1.8 | |
| 19 | Technical infrastructure (e.g., power transformation areas) | 0.4 | 0.5 | 0.6 | |
| 20 | Terrace houses | 1.1 | 1.7 | 1.0 | |
| 21 | Urban forest of considerable size and compactness | 2.0 | 4.5 | 4.7 | 1 |
| 22 | Villas | 1.5 | 2.1 | 1.4 | |

## 2.3. Impact Assessment Using a Standard Approach

We analyzed the potential impacts of each of the four access roads using a parallel assessment. First, the impact assessment focused on three fundamental environmental aspects to estimate the necessary compensation, based on the protected environmental goods in German legislation: soil, biotope and recreation. We defined a buffer area of 50 m for each of the proposed access roads. We measured the high-quality soil, biotope and recreation values that would eventually be lost within each of these buffer areas.

### 2.3.1. Biotope

A biotope is evaluated based on vegetation cover from the perspective of nature conservation (biotope value). The value of biotopes refers to the scheme to manage compensation used in landscape planning [32]. Biotope value depends on the degree of naturalness, rareness, recoverability, and integrity. Biotope value is expressed on an ordinal scale from 0 (the lowest) to 10 (the highest). Our assessment focused only on the highest occurring biotope values (6–7 high and 8–9 very high), for which we calculated the respective hectares. We used the detailed biotope map provided by the cities of Bochum and Dortmund. The maps were prepared at a scale of 1:5000 and are regularly updated. We manually determined the values of empty areas by identifying equivalent biotopes.

### 2.3.2. Soil Values

Soil map units include soil quality based on a long-established assessment scheme [33], which assigns numbers according to a soil's relative capacity to bear and sustain crop production. Soil quality is assessed on the basis of the soil texture, which reflects a soil's

capacity to store plant available water and nutrients; soil parent material and soil development also reflect natural nutrient provisioning. Climate was also considered. All aspects were combined in a dimensionless, ordinally scaled indicator to express the relative differences in net agricultural yield from 1 to 100. Generally, a soil map assigns one of five land value classes to each soil unit (very low, low, medium, high, and very high). The class, very low, did not occur in our area. The analysis concentrated on measuring the hectares lost in areas with medium and high soil values in open spaces, as there were no very high values areas in the impacted areas. We evaluated soil only in open spaces because these areas have not been sealed, and, in the case of constructing the road, such soil would eventually be lost.

### 2.3.3. Recreation Values

To estimate the recreational value of a particular USS, we used the assessment of cultural ES, as recreation is a cultural ES. We used the values obtained in an expert workshop with 11 scientists and professionals, as described in Section 2.4.1. We selected the five C-ES directly connected to recreation (column R in Table 2); therefore, this recreational value was slightly smaller than the estimated cultural ES. To estimate the recreational value, the analysis calculated the area loss only in terms of the USSs considered open space (column p in Table 1).

### 2.4. Impact Assessment Using ES

In the second step, we applied the ES land cover matrix and expert assessment approach [14,34,35] to evaluate the impacts of alternatives and to identify potential areas for compensation measures. The spatial scope of the compensation measures was restricted to the analyzed area for which the identification and assessment of ES were performed.

### 2.4.1. Mapping ES

For the identification of relevant ES, we relied on a workshop with 11 experts in planning and science working in the Ruhr area. Using CICES v5.1 [11], the experts identified the 25 most relevant ES for the study area (Table 2). Then, the expert panel assessed the potential ES supply of each USS (Table 1) using a scale from 0 to 5 (null to very high). As the evaluations of the experts slightly differed, we averaged the experts' scores. To calculate the respective bundle, we average the single ES values of the ten provisioning (P-ES), nine regulating (R-ES), and six cultural ES. Using these bundle values, we mapped the bundles of ES following the matrix approach [3,14,15,36]. The full list of identified ES, with their respective CICES codes, is presented in Table 2.

We calculated the total ES supply within the 50 m buffer area for the four analyzed access roads using Equation (1):

$$E_k = \sum_{i=1}^{n} (a_i e_i) \tag{1}$$

where:

$E_k$ = total ES supply in buffer $k$ (P-ES, R-ES, or C-ES)
$a_i$ = Surface of USS $i$ present in the buffer area $k$
$e_i$ = Supply of ES of USS $i$ (according to Table 1).

### 2.4.2. Identifying Hot and Cold Spots of Supply

To identify the areas with the highest and lowest supplies of ES, we mapped the hot and cold spots for P-ES, R-ES and C-ES at 90%, 95% and 99% confidence levels using the Getis-Ord Gi* tool in ArcGIS 10.1 ©. Using this method, we ensured a spatially explicit and meaningful identification of areas containing clusters of high and low ES supplies. These areas were also used in the design of compensation. The analysis of hot–cold spots was performed over a hexagonal grid of 1 ha.

**Table 2.** Selected ES used in the assessment. Column R indicates the ES used to estimate the recreation value.

| Bundle | | Class | CICES Code | R |
|---|---|---|---|---|
| 1 | | Cultivated terrestrial plants (including fungi and algae) grown for nutritional purposes | 1.1.1.1 | |
| 2 | | Animals reared for nutritional purposes | 1.1.3.1 | |
| 3 | Provisioning (Biotic) | Animals reared by in situ aquaculture for nutritional purposes | 1.1.4.1 | |
| 4 | | Wild plants (terrestrial and aquatic, including fungi and algae) used for nutrition | 1.1.5.1 | |
| 5 | | Wild animals (terrestrial and aquatic) used for nutritional purposes | 1.1.6.1 | |
| 6 | | Surface water used as a material (non-drinking purposes) | 4.2.1.2 | |
| 7 | | Freshwater surface water used as an energy source | 4.2.1.3 | |
| 8 | Provisioning (Abiotic) | Ground (and subsurface) water for drinking | 4.2.2.1 | |
| 9 | | Ground water (and subsurface) used as a material (non-drinking purposes) | 4.2.2.2 | |
| 10 | | Ground water (and subsurface) used as an energy source | 4.2.2.3 | |
| 11 | | Filtration/sequestration/storage/accumulation by micro-organisms, algae, plants, and animals | 2.1.1.2 | |
| 12 | | Noise attenuation | 2.1.2.2 | |
| 13 | | Visual screening | 2.1.2.3 | |
| 14 | | Hydrological cycle and water flow regulation (Including flood control) | 2.2.1.3 | |
| 15 | Regulation & Maintenance (Biotic) | Pollination | 2.2.2.1 | |
| 16 | | Maintaining nursery populations and habitats (Including gene pool protection) | 2.2.2.3 | |
| 17 | | Decomposition and fixing processes and their effect on soil quality | 2.2.4.2 | |
| 18 | | Regulation of the chemical condition of freshwater by living processes | 2.2.5.1 | |
| 19 | | Regulation of temperature and humidity, including ventilation and transpiration | 2.2.6.2 | |
| 20 | | Characteristics of living systems that that enable activities promoting health, recuperation or enjoyment through active or immersive interactions | 3.1.1.1 | R |
| 21 | | Characteristics of living systems that enable activities promoting health, recuperation or enjoyment through passive or observational interactions | 3.1.1.2 | R |
| 22 | Cultural (Biotic) | Characteristics of living systems that enable scientific investigation or the creation of traditional ecological knowledge | 3.1.2.1 | |
| 23 | | Characteristics of living systems that enable education and training | 3.1.2.2 | R |
| 24 | | Characteristics of living systems that enable aesthetic experiences | 3.1.2.4 | R |
| 25 | | Elements of living systems that have symbolic meaning | 3.2.1.1 | R |

*2.5. Evaluation of Compensation*

We used the direct loss per buffer calculated with Equation (1) ($E_k$) as a value for ES compensation. To evaluate the possible compensation for ES loss, our criteria were twofold: (1) to maintain the same amount of ES supply existing in the selected buffer

and that redistributed in a sector located near the buffer area, and (2) to maintain the same amount of open space that will eventually be lost; this surface will be relocated in the nearby impacted area. To identify the area for compensation, we used the analysis of hot–cold spots to select the cold spot close to the selected access road. We performed the calculations for compensation using a hexagonal grid of 1 ha. Using a grid approach helped to understand the analyzed impacts that were more in line with aspects of urban form, as, normally, LULC units are hierarchically arranged in space and time following six fundamental aspects [37,38]. The hexagonal grid is a powerful tool to depict the spatial structure of urban environments [39], because it can summarise the high heterogeneity the land uses that occur over short distances in a comparable manner.

For the estimation of ES supply for each hexagonal cell, we used the same Equation (1) used for the calculation within buffers. The total amount of ES to be compensated in the new area corresponded to the total amount of ES in the respective buffer ($E_k$), an assumption that only maintains the current situation—no quantitative ES improvement. This ES total amount was the sum product of the ES (P-ES, R-ES and C-ES) and the respective surface in hectares covered by each of the USSs within the buffer. To estimate the amount of ES to be compensated per cell, we used Equation (2):

$$E_c = \frac{E_k - \left(\sum_{i=1}^{p}(e_i - e_h)\right)}{(m + |p|)} \tag{2}$$

where:

$E_c$ = ES to be compensated in cell $h$ (P-ES, R-ES, or C-ES)
$e_h$ = existing ES supply in cell h (calculated with Equation (1))
$E_k$ = total supply of ES in buffer $k$ (P-ES, R-ES and C-ES respectively
$e_i$ = ES supply of open space USS (Table 2, P-ES, R-ES, C-ES where $p$ = 1)
$m$ = number of cells in the cluster to be compensated
$p$ = number of cells to be replaced with open space, as shown in the following:

$$p = |p_k| \tag{3}$$

$p_k$ = number of open space USS (Table 1), and
$k \in Z: 1 \leq i \leq 5$.

Then, the new ES value per cell to be mapped was:

$$E_t = E_c + e_h \tag{4}$$

## 3. Results

### 3.1. Land-Use Change Impact

The study area contained a core of residential and industrial uses surrounded by open spaces. Massive railroad infrastructure running from East to West separates the area into two parts. The three predominant USSs were "housing complexes with green areas", e.g., row houses and single multi-story houses, "arable fields", and "urban forest", covering 17.8%, 13.8% and 12% of the total area, respectively. Open space covered 36% of the total area at approximately 591 ha (see Figure 2).

Due to the particular spatial distribution of the USS and the differences in the surfaces covered by each of the proposed access roads, the potential impacts in terms of land-use change varied. Access A1 will affect 25.2 ha, of which 3.8 ha (15%) corresponds to open space. In the case of A2, the area impacted will be 25.3 ha, containing 7.7 ha (30%) of open space. Access A3 will affect 16.6 ha and 8.9 ha (53%) of open space. Access A4 will affect 15.8 ha, and the open space within that area is 6.4 ha (40%). In terms of absolute open space impact, alternative A1 has the lowest impact, followed by alternatives A4 and A2, while alternative A3 has the highest impact (see Figure 3).

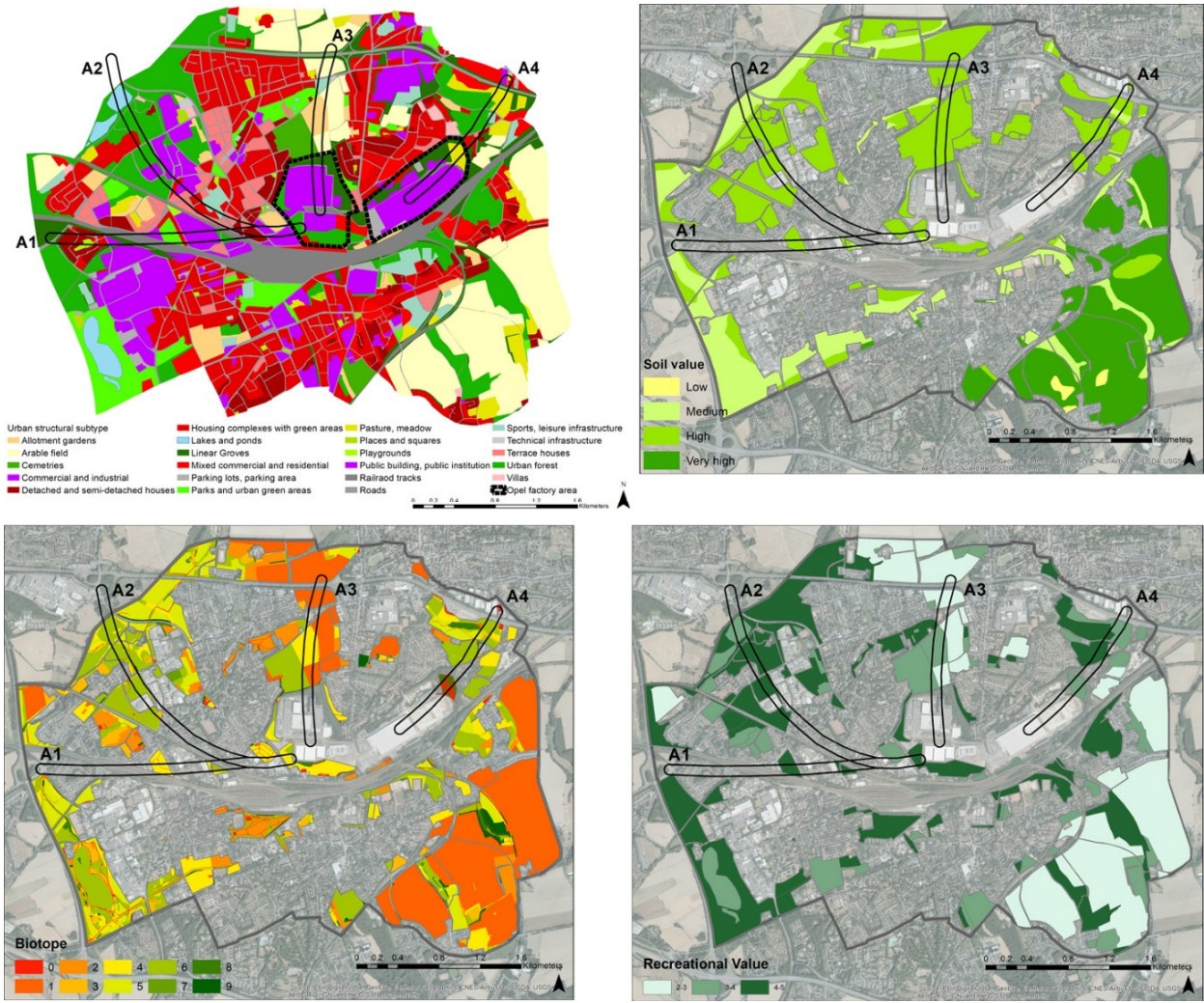

**Figure 2.** Urban structural subtypes in the study area (**upper left**). The areas of the European logistics centre for the distribution of spare car parts and the new industrial area are indicated with a black segmented line. The 50 m buffers of the four access roads are designated by the abbreviations A1, A2, A3 and A4. These buffers were used to calculate losses in the specific impacted areas. Soil values in open space (**upper right**). Biotopes values in open space (**down left**). Recreational values in open space (**down right**). Source: own elaboration and the assessment is based on information provided by the city of Bochum 2017 and Geological Survey NRW; satellite image as in Figure 1.

The USS most impacted by A1 was "commercial and industrial use", accounting for 5.6 ha and 22.4% of the total impacted area. In the case of A2, the USS most impacted was "urban forest", with 7.2 ha (28.3%) affected. A3 impacted 4.2 ha (25.5%) of "arable fields", while A4 also impacted "commercial and industrial uses" at a similar amount as that of A1 at 5.6 ha (33.6%). In terms of the impact on the USSs classified as open space, A2 had the highest impact due to the 7.2 ha of urban forest affected. The second-highest impact occurred with A3, due to the impact on "allotment gardens" and "arable fields". A4 impacted 3.6 ha of pastures and meadows, and, finally, A1 had less of an impact, with 1.7 ha of "urban forest" and 0.5 ha of "allotment gardens" affected.

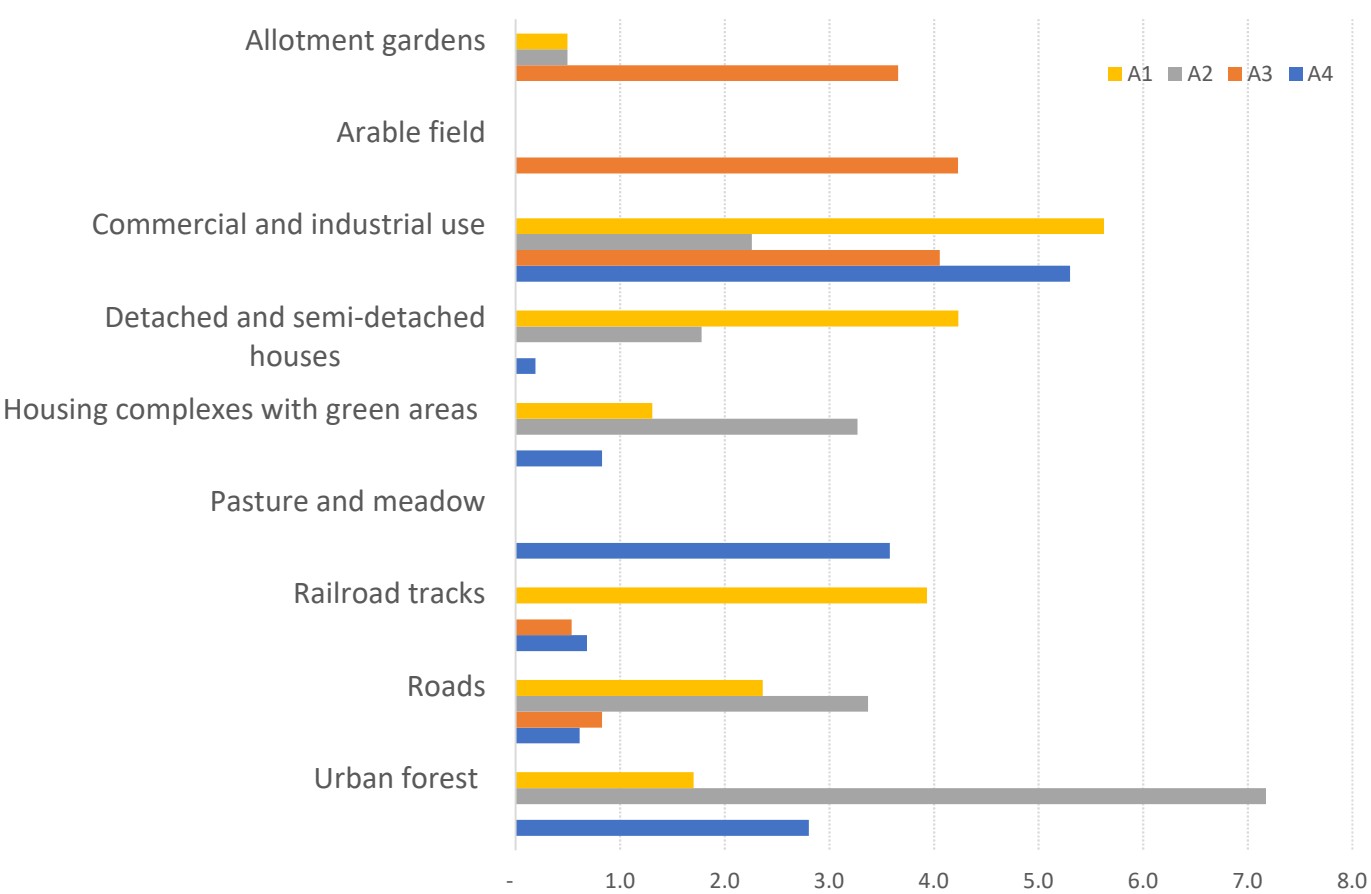

**Figure 3.** The USSs area lost from the development each of the access roads. Only the first three larger USSs in each alternative were plotted.

Impacts on Soil, Biotope and Recreation

When the assessment included the quality of the impacted areas in terms of soil, biotope and recreation, the situation was similar to that described above. Figure 4 shows the highest and the lowest impacts of the access roads, according to the respective losses in terms of hectares, of high-quality soil, biotope and recreation. The most severe losses of good quality soil, biotope, and recreation were connected to A2, which had substantial impacts. The lowest impact for the three analyzed variables was found again in A1. High-quality soil will be impacted most by A3.

### 3.2. Impacts on ES

In this section, we assess the impacts on ES, quantifying the impact of each of the access options in terms of P-ES, R-ES and C-ES (Figure 5). Similar to the previous analysis, A2 had the highest impact. However, the situation regarding the lowest impact changed, with A4 having the lowest impact. Considering the impacts in terms of ES bundles, A4 showed the lowest impact for P-ES and the second-lowest impact for R-ES and C-ES.

### 3.3. Analysis of Compensation Areas using ES

The spatially explicit assessment of ES using the hexagonal grid in the whole study area is presented in Figure 6, in addition to the analysis of hot and cold spots for provisioning (P-ES), regulating (R-ES), and cultural (C-ES) services. The results show a strong contrast between densely settled and industrial areas that produce cold spots, with the lowest ES values in the core, and the open space at the outskirts that concentrate the hot spots with the highest ES values.

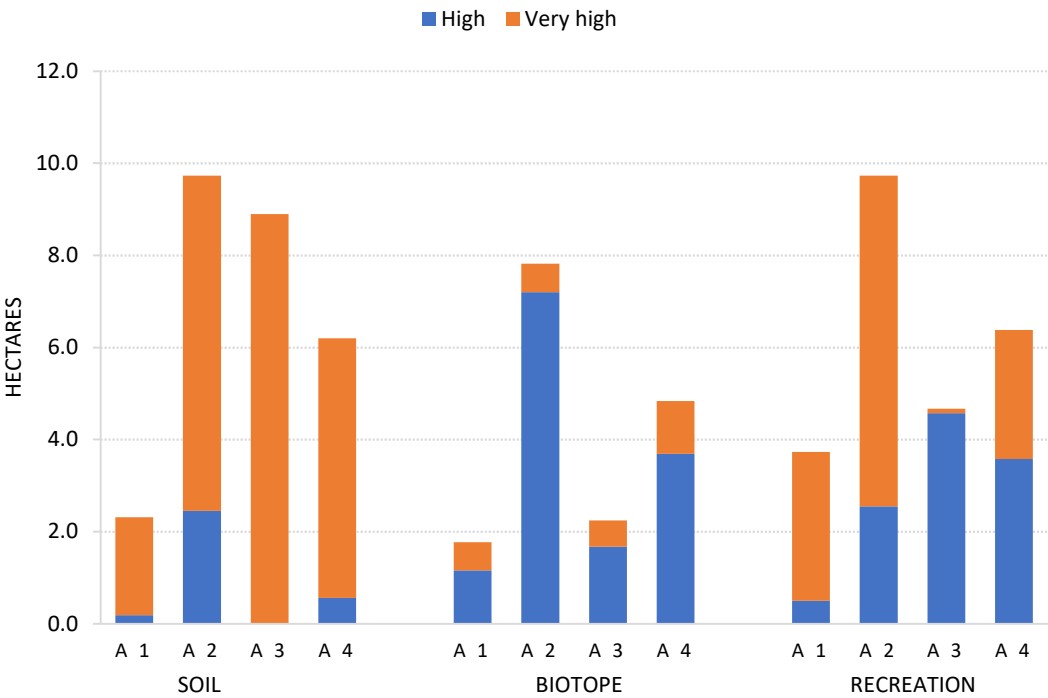

**Figure 4.** Area losses (hectares) of soil, biotope, and recreation for the four access road variants. The analysis considered only the areas providing high and very high values for each of the selected variables. In the case of soil, medium and high values were considered.

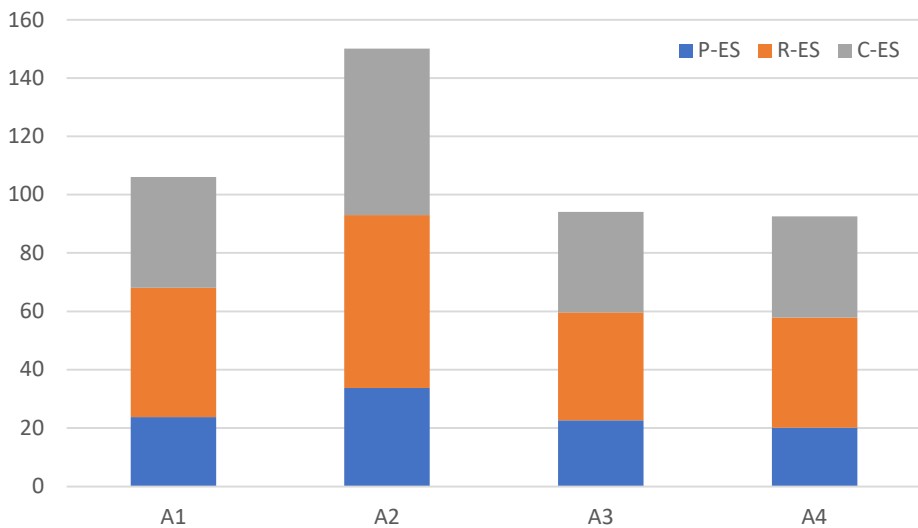

**Figure 5.** Impact of access roads in terms of ES. The impacts in terms of R-ES and C-ES for A4 are slightly higher than those for A3. Values for the P-ES, R-ES and C-ES were calculated using Equation (1) ($E_k$).

To illustrate our scheme of on-site compensation measures, we used A4, the alternative with less impact in terms of ES, according to our previous analysis. As a compensation area, we selected the hexagons present in a C-ES cold-spot cluster directly impacted by A4 (Figure 6). This cluster contained 63 hexagonal cells. We evenly distributed the amount of ES losses within these 63 cells. The amount to be compensated corresponded to the total sum product of the ES values per the USSs, P-ES, R-ES and C-ES, which are presented in the table in Figure 5. To fulfil the requirement of no net loss of important open space, we considered the replacement of the existing 3.6 ha of "pastures and meadows" and the 2.8 ha of "urban forest" that corresponded to the 6.4 ha of lost open space contained in

the A4 buffer (Figures 3 and 5). Each hexagonal cell was 1 ha, which corresponded to four complete cells of a new urban forest, and three new cells of pastures and meadows. We discounted the supply of the new urban forest and pasture and meadow cells from the total ES amount to be compensated. The remainder was evenly distributed within the 56 target cells.

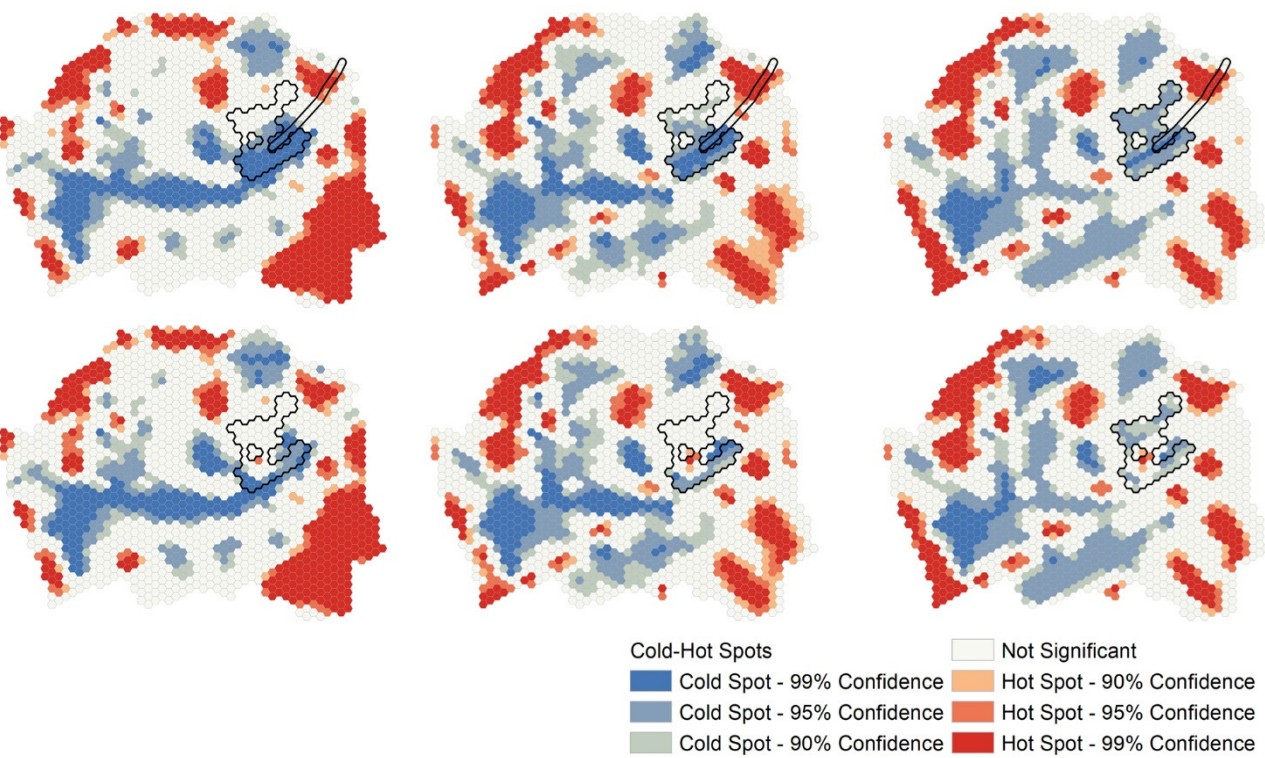

**Figure 6.** Compensation example for A4. Spatial distribution of ES. Upper row: total supply of provisioning ES (**left**), regulating ES (**center**), and cultural ES (**right**); center and lower rows: respective analyses of cold–hot spot for the current situation (**center**) and improvements after compensation (**lower**). The selected compensation area is indicated in black.

The central row in Figure 5 shows the analysis of cold–hot spot without compensation. The spatial structure of the ES describes a doughnut effect, with the outer areas forming a ring of hot spots and the inner areas containing cold spots. In the previous step, we selected alternative A4, which provided the amount of ES to be compensated. Once this compensation of ES was added to the selected cells, the existing cold spot was severely reduced. In the case of R-ES and C-ES, a new, small hotspot was generated in the area. This new small hot spot means that the improvement in the ES supply in the impacted area was, therefore, substantial.

## 4. Discussion

To be meaningful for society and decision-making, ES assessments should avoid the impulse to "distil the value of nature into a number (monetary or otherwise), and then communicate that number broadly" [40]. Alternative approaches to a single monetary value are better for considering what people care about in terms of specific decisions at stake, thus linking ES with social considerations. While ES science has evolved in meaningful ways to fulfil the promise of supporting decision-making, it is necessary to advance better characterizations of ES change, coupled with multimetric and qualitative context-specific valuations [40].

ES mapping has been increasingly used to assess and predict the expected impacts of urban development as a way to increase the quality of planning decisions [25,41]. The paradox of urban development is that attempts to increase the quality of urban environments can, at the same time, harm ES by sealing soil, fragmenting habitats, losing open

space, and diminishing important vegetation structures, and, therefore, threaten human well-being [25,42].

Our empirical, methodological approach can counteract urban development short-comings in a sound environmental compensation manner that accounts for the losses of ES. The proposed consideration of the spatial distribution of ES in a wider area surrounding a site that is affected by land-use change broadens the perspective. Our assessment gives an overall picture of an area's environmental situation, allowing multiple possible analyses of environmental performances, risks and strengths. We illustrated only three environmental aspects: soil, biotope and recreation. Similar studies have proven ES mapping a powerful input for scenario evaluation [3]. Here, we demonstrated that ES mapping using expert assessment can assist in evaluating possible impacts arising from urban development and in analyzing compensation. We presented two ways to analyze the impacts of urban development, with contrasting results. One way was based on the area-weighted loss of protected environmental goods, and the other way highlighted the impacts on ES. The merit of an ES assessment, as analyzed in the introduction (Section 1.2), is that it integrates into the decision-making process the hidden or underrepresented values of nature that could eventually be lost. Our analysis also included the full spectrum of USSs and the respective benefits that society receives from ecosystems in such areas. An analysis based on expert assessments of ES and the LULC matrix is highly correlated with biophysical measurements [16]. Therefore, our assessment is suitable for rapid preliminary evaluations and can be complemented or improved with biophysical measurements of key ES for a more complex analysis.

### 4.1. Limitations

There are two general limitations to consider when using our approach. The first limitation is that we used a general delineation of the planned roads. A more detailed assessment should use the exact delineation as detailed in engineering plans. The second limitation is that we did not assess in detail the area compensated in terms of USSs. In general, it is always possible to increase the supply of ES by including NbS in existing urban areas. However, a detailed estimation is necessary according to specific urban morphology.

### 4.2. The Role of Participatory Processes

Public participation can be greatly supported by using ES as a basis for guiding discussions and focusing on synergies and trade-offs between development options and stakeholder groups [1]. Our method allows the integration of civil society, a key aspect of participatory planning [1]. In the participatory process held in urban planning, argumentative lock-in-situations can block sustainable urban development. Environmentalists and environmental authorities remain in fierce opposition to municipal development agencies and developers. The former focuses on protection; the latter favours using open space for urban development while offering new economic possibilities and jobs. As a consequence, and due to powerful lobbies, city councils often overrule environmental concerns. Consequently, there is progress for one group of stakeholders only, at the expense of the other groups. We illustrated the procedure with an actual case study, with results that can feed a participatory process. Our method allows the use of participatory planning, where stakeholders and experts can identify development alternatives that could be judged by citizens assisting in a structured decision-making process that is value-focused, i.e., able to express what matters to people, and analytic, i.e., ascertains trade-offs between alternatives, as suggested by Chan [40]. Such an approach, where ES "values drive alternative scenarios and spatial analysis of benefits and costs" [40] holds the potential for reducing the social impacts brought on by urban development [1]. The theoretical basis of the ES framework is robust, but to put it into practice requires the involvement of key agents of development [43].

*4.3. Environmental Encroachment Paradox*

The assessment of environmental impacts induced by land-use change, such as the construction of road infrastructure, requires the inclusion of neighbouring areas that are spatially and functionally related to the directly affected site, and the taking into account of the urban context of urban transformations at larger scales. When ES are not assessed in areas where urban development takes place, the supply of ES might be reduced, affecting the well-being of residents. This local loss of ES can occur even if the overall ES supply remains equal or has been improved at larger scales.

The traditional approach of environmental impact assessments is to assess and judge possible encroachment on the environment expected at specific development sites. Mandatory compensation for the loss of nature (e.g., according to Directive 2011/92/EU) [44] is often restricted to replacing urban green spaces by improving rural or peri-urban green spaces, as in the case of German legislation [45]. According to the current legislation in Germany, compensation is restricted to the encroachment of legally protected goods (people and especially human health, animals, plants, biodiversity, soil, water, air, climate and landscape, cultural heritage and other properties, as well as interactions between protected goods) and must be carried out by balancing the loss of biotopes by increasing the quality of biotopes in other places. Typical compensation measures are planting trees, creating artificial wetlands, and transforming arable fields into species-rich meadows, or simply compensating by setting aside money for nature conservation purposes without any direct localised effect. These sites do not necessarily need to be close to the area of environmental encroachment. At the moment, even eco-accounting between distant locations and monetary compensation is legally possible. The highly valued vegetation will be placed far from the location of encroachment, making the compensation measures from such assessments lead to an upgrade in the quality of already existing open and green spaces elsewhere. Consequently, the environmental quality of the directly affected people, those living in the affected area, would not profit from such compensation. Thus, benefits may not be noticeable on or near the site impacted by the land-use change. Eventually, shifting compensation in distant areas means a deterioration of the supply of ES in the affected area.

Such legally binding, mandatory compensation for the loss of nature has several shortcomings: (i) compensation is carried out by replacing vegetation with vegetation; (ii) the loss of ES in areas to be developed is neglected; and (iii) the current environmental status is not improved. Vegetation and protection of species serve as the main indicators of the value of green elements. The degree of compensation is usually calculated by multiplying the biotope value with the area of the lost urban green space, neglecting other ES; and (iv) in practice, compensation is allowed to occur in distant areas, far from the affected site. In contrast, we suggest that compensation should include measures that maintain or eventually improve the environmental and living conditions in the affected area. Such measures could include countermeasures against urban heat islands, the realisation of nature-based solutions (NbS) against urban flash floods, reclamation of brownfields, river restoration, and other NbS. Using an ES assessment, it is possible to improve existing environmental quality in the affected areas. Our method goes beyond the calculation of impacts for each variant. We show how urban development can contribute to improving the overall environmental situation of areas subject to development, using the ES framework.

The consideration of the broad spectrum of NbS, along with ES supply, can open a vast range of possibilities for compensation. Admittedly, regulations that define compensation measures might have to be diversified, and practice in municipal administrations and consultancies needs to be adapted, for the ES concept to unfold its full advantage.

*4.4. Estimating Urban Development Compensations with ES*

To apply the ES framework in urban settings, it is necessary to include all types of urban areas belonging to the urban ecosystem, to consider the full spectrum of land-uses and not restrict the analysis only to the urban green space, as normally occurs [37]. Dif-

ferent degrees of urbanisation will deliver ES at different intensities, which can be higher for some bundles, such as cultural services, or lower for others, such as provisioning services [12,31,36]. Restricting the analyses only to green areas is a conceptual and methodological shortcoming which constrains the chances to ameliorate development alternatives in the places where it is needed. Furthermore, combining compensation measure determination with an ES assessment can improve preexisting situations, as we have indicated with our analysis.

Our approach ensures that the supply of ES will be maintained in the impacted areas, considering the broad spectrum of USSs. This means that the benefits of nature can be found everywhere. The relevance lies in the amount of ES—to maintain or increase when the amount is too low. Nature is not restricted to open space.

In our approach we considered no net loss of ES, i.e., maintaining the existing supply. It is also possible to apply this procedure to improve an existing situation. This could be realized by introducing a coefficient defining a target for ES improvement in the specific compensation area, or by targeting specific cold spots to transform them into hot spots.

*4.5. ES and Urban Form: Spatial Distribution Matters*

Enhancing the supply of ES in urban environments involves aspects of spatial distribution. In our approach, we maintained the same amount of preexisting ES that had been relocated to a nearby area previously identified as a cold spot for cultural ES. The analysis of cold and hot spots after the calculation of the compensation showed that the former cold spot was diminished in size, and partially transformed into a hot spot. This outcome indicated a substantial change in the spatial structure of the ES supply. It also showed that the supply of ES can be enhanced, even when no explicit quantitative improvements are considered, because a spatial structural assessment was involved in the supply of ES. Our results are in line with those in a similar study by Thomas [46], which showed that only by changing the spatial distribution of ES supply areas can the overall supply be altered, enhanced or diminished.

## 5. Conclusions

Our findings are relevant to research on the spatial structure of urban ES and the changes introduced by urban development. The peculiarities of the spatial structures found in urban areas, high heterogeneity, and complexity over relatively short distances, make the urban form a fundamental aspect to consider within ES assessments. Using USSs and a matrix-based approach for the assessment of ES allows the incorporation of the urban spatial structure that underpins the supply of ES. In our assessment, we illustrated how the selection of road alternatives can be improved by using an ES framework.

Our method illustrates how to improve planning procedures using an ES based approach. The suggested methodology is meant to counteract some of the shortcomings of traditional, legally binding regulations for environmental compensation due to urban development. Our method aims to strengthen urban resilience with a holistic understanding, using ES. The advantages of our method are the following: (i) not restricting compensation to merely accounting biotopes, but considering a wide range of ES; (ii) taking into account the overall spatial distribution of ES in the affected area in the search of legally mandatory compensation; (iii) highlighting synergies and trade-offs and allowing for NbS and strategic implementation of green infrastructure measures to leverage co-benefits [47]; and (iv) providing compensation near the affected locations.

**Author Contributions:** Conceptualization, H.Z. and L.I.; data curation, H.Z. and L.I.; formal analysis, L.I.; funding acquisition, H.Z. and L.I.; investigation, H.Z. and L.I.; methodology, H.Z. and L.I.; visualization, H.Z. and L.I.; writing—original draft, H.Z. and L.I.; writing—review and editing, H.Z. and L.I. All authors have read and agreed to the published version of the manuscript.

**Funding:** The finalization of this paper was supported by the research project, "Implementation of the Concept of Ecosystem Services in the Planning of Green Infrastructure to Strengthen the Resilience of

the Metropolis Ruhr and Shanghai—research grant 01LE1805A", funded by the Bundesministerium für Bildung und Forschung (BMBF).

**Data Availability Statement:** Data available on request. The data presented in this study are available on request from the corresponding author.

**Acknowledgments:** Mapping was partly carried out in a transformation lab within the framework of the Double Degree Master Program, "Transformation of Urban Landscapes", jointly offered by Ruhr-Universität Bochum and Tongji-University, Shanghai.

**Conflicts of Interest:** The authors declare no conflict of interest.

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
