# Peer review of "Who Pays the Bill? Assessing Ecosystem Services Losses in an Urban Planning Context"

_land, doi:10.3390/land10040369_

Round 1

Reviewer 1 Report

Overall, a nice paper that provides a nice example of how an ES analysis can be used for a very specific local scale decision scenario (building road), including how this analysis compares to other types of approaches focused on environmental quality.

I noted some places where additional clarification is needed. My primary recommendation is to add more discussion of why the results of the standard analysis differed from that of the ES analysis, in particular including 1) how the approach to measurement (hectares of loss vs. overall score) may have affected interpretation between the two analyses, 2) how the more data-based approach of the standard analysis vs. the expert judgment approach of the ES analysis may have affected interpretation. Also, the methods describe an ES compensation analysis that is only presented as an example for one decision scenario (A4), but I think it would be worthwhile to see the results of this compensation analysis for the other scenarios as well.

Title – “Who pays the bill?” – I think this title is misleading.  There is really no discussion of who will pay the bill or costs of implementation, and that is not really what the analysis is about;

Line 95 – “An ES framework can…” or “The ES framework” or “ES frameworks can…” or similar

Line 99 – “enhancing” instead of “enhance” to match “supporting” and “addressing”

Line 108 – incorporate them into what?

Line 119 – “Avoidance” with a capital A to match the other numbers

Line 217 – Since the use of the ES expert scores comes Before you describe how they were done, I suggest adding something like “as described in Section 2.4” or similar just to be clear

Line 233- Just to be clear – the expert panel assigned a value from 0-10 for each of the 25 ES for each USS  (i.e., a total of 25 x 22 scores), which were then averaged by USS to create a mean (?) score for each of P-ES, R-ES, and C-ES?  Did the panel assign a single score, or was each panel member interviewed separately to generate an average score? ETc.   Just a few more details on exactly how the scoring was done would be helpful.

Figure 2 – in the two lower graphs I think “Biotop” should be replaced with “Biotope” and “Recreational value”

Line 315 – this technical recommendation (based on minimizing open space) doesn’t seem to consider other factors when building a road (such as the fact that open space may be the only available property on which to build the road without having to tear down existing infrastructure); A1 seems to run through public buildings and housing; is preserving open space really a priority over preserving existing infrastructure? I think this technical recommendation needs to be more carefully worded here, and this broader issue of tradeoffs between ES and other factors included in the Discussion

Figure 3 – If the table and graph contain identical information, you really don’t need both; there is a typo in “railroad”; the x-axis on the graph needs a label and units; you could also consistently capitalize the USS labels; also, why are these plotted in reverse order (both alphabetical of USS and the order of A4-A1)

Line 323 – I’m not sure what is meant by “in terms of environmental quality” – I think you are saying urban forest is a higher quality USS than commercial use or arable field, but is this just a general assumption or is it based on the soil/biotope/recreation analysis in 3.1.1. Either clarify what is meant by quality here, or I think wait to discuss quality until the next section?

Line 333 – biotope and recreation are mentioned twice in this sentence;

Figure 4 – I know soil values were categorized as low to very high, but it is not clear exactly what is meant by high or very high biotope or recreational values – what was the cutoff to be considered high or very high?

Figure 4 – Again, if the table and the figure are identical here, you don’t need both; suggest making labeling consistent with Figure 1 and the main text (A1 instead of Access 1); you also don’t necessarily need the numbers overlaid on the figure bars, as it is fairly easy to interpret the hectare numbers from the graph

Line 346 – it is an oversimplification I think to say “A3 is a worse option” – you are basing this on Open space hectares. If for example, the priority was to preserve Regulating ES or Cultural ES, then A3 might seem like a better option.  It depends on the tradeoffs across R, P, C and open space.

Line 347 – “more of open space lost” – by talking about open space, you seem to be mixing your first analysis (open space, biotope, soil, recreation) with your ES analysis; I think it would be better to focus on just the ES results here, and then bring in open space as part of the final compensation synthesis (as described in Line 261);  Otherwise you should clarify in you methods that your standard analysis and ES analysis include each’s respective metrics + open space;

Figure 5 – Again if table and graph are the same, you don’t need both; you need a y-axis label and units; What are the units on the ES – those are total scores (Ek)?  What is the “average” of – just the ES I assume, but as written it appears to also include Open space;  I wonder if it is necessary to include Open space as part of this figure/table – it was already discussed in Lines 309-315 and is just repeated here;

Line 361 – does the 63 cells of the compensation area include the road location? Based on Fig. 2 the road appears to go through the compensation area, but it is difficult to tell

Figure 6 – You should give ranges on what is meant by very low to very high – i.e., are these based on quintiles or just a uniform division of scores (0-20, 20-40, etc.); These are essentially the Ek scores in each hexagon, right?  You should include in the Figure legend that this is the example for A4.

Figure 6 – It is difficult to visualize how these hexagonal grids overlay on the maps in Figure 2.  Could the road A4 be added? From what I can tell, the road runs through the compensation area?  But the text says the compensation area was selected near the road? I am trying to compare the bottom row to the middle row to see exactly where ecosystem services were lost and where they were compensated for, but it is difficult without visualizing the road location.

Section 3.3. – The way this section is written, it appears you only did the compensation analysis for option A4, as none of the other options are discussed. Could the A1, A2, A3 options be added to Figure 6, so that the reader could see how these options change the landscape?  I realize this would only work if the compensation areas do not overlap;  to save space, I think you could eliminate the top row (the black and white ES scores) as it is somewhat redundant with the middle row;

Line 386 – “avoid the temptation to distill into a single number” – although you kind of do this by distilling your 25 ES down to a single ES score, rather than looking at tradeoffs among them individually.  This point is illustrated in line 346 where you declare A4 the winner based on overall ES score (even though A3 wins for both R-ES and C-ES);  If, for example, the community prioritized C-ES more strongly than P-ES, they might prefer A3.  Furthermore, by consolidating your 25 ES metrics to 3 categories, you also potentially lost a lot of information on tradeoffs, assuming all 25 metrics are of equal importance. It is worth a consideration of what was lost or gained by taking this approach (rather than examining metrics individually).

Line 387 – typo – capitalize “It”;  Actually the phrasing “it is required” seems too strong;  I think you mean an alternative (better) approach to a single monetary value is a broader consideration multiple ES linked with social considerations; This could be more clearly stated

Line 411 – “with contrasting results” – I think there merits further explanation here as to why A4 is the best option in the second analysis, but A1 is in the first;  What additional information is the ES analysis providing that caused the changed in results; 

For example, Recreation in the first analysis and C-ES second analysis are calculated very similarly with the exception of 1 ES (scientific investigation and knowledge);  So I’d expect the results in Figure 4 and Figure 5 to be fairly similar – yet in Fig. 5 A1 has higher recreational value (A2>A1>A4>A3), whereas in Fig. 4 A4 does (A2>A4>A3>A1).  Why the difference?   

Overall the ES analysis has more metrics – so is it possible that the metrics (other than those related to Soil or Biotope) tended to be lower for A4, so as more metrics are included, then A4 starts to look like a better option? In other words, is it really the inclusion of more types of metrics driving the pattern (broad focus on 'provisioning' vs. narrow focus on soil?)? 

Alternatively, the results of the Standard analysis and ES analysis are based on different measures (hectares of high or very high quality, vs. total ES score) – had the ES analysis been based on hectares of high/very high as well would the results have changed?  Is the inclusion of lower quality ES "hectares" dragging down the A1 score in the 2nd analysis?

To what extent do you think expert opinion to develop the ES scores drove the results (more or less accurately) than the approach to quantifying soil or biotope (which were more data based rather than expert opinion)?  Could the observed differences between the two analysis reflect uncertainty in how the values are measured?

Etcetera – bottom line I think it is worth more discussion about how/why the approach to the analysis caused the ‘preferred’ solution to change

Author Response

The authors would like to express their gratefulness for the helpful comments and suggestions made by reviewers to improve the paper. We have taken all remarks carefully into consideration and responded to them accordingly.

Reviewer #1

Comment

Response

Overall, a nice paper that provides a nice example of how an ES analysis can be used for a very specific local scale decision scenario (building road), including how this analysis compares to other types of approaches focused on environmental quality.

Thank you very much for your comment.

I noted some places where additional clarification is needed. My primary recommendation is to add more discussion of why the results of the standard analysis differed from that of the ES analysis, in particular including 1) how the approach to measurement (hectares of loss vs. overall score) may have affected interpretation between the two analyses, 2) how the more data-based approach of the standard analysis vs. the expert judgment approach of the ES analysis may have affected interpretation.

WE have included more discussions:

1)      Regarding the point 1,

2)      Regarding point 2,

Also, the methods describe an ES compensation analysis that is only presented as an example for one decision scenario (A4), but I think it would be worthwhile to see the results of this compensation analysis for the other scenarios as well.

We have included the alternative scenario A4 because is the one having the best performance in terms of ES. Our idea was to illustrate how compensations can be made, once alternatives are selected. It falls out of the scope to include the compensation of all alternatives. That is also something which is not done in real projects, where compensations are evaluated only for selected alternatives. Not to mention the massive amount of GIS processing that this suggestion involves, it will dilute our approach by adding unnecessary analysis.

Title – “Who pays the bill?” – I think this title is misleading.  There is really no discussion of who will pay the bill or costs of implementation, and that is not really what the analysis is about;

We believe that the title is correct. The bill will be always paid by local residents. They are the ones suffering environmental encroachment and loss of ES, which could eventually and theoretically be expressed in monetary units (cf. the TEEB approach). To this end, we have developed this approach to account for the ES that need to be compensated.

Line 95 – “An ES framework can…” or “The ES framework” or “ES frameworks can…” or similar

Done. Sentence was changed.

Line 99 – “enhancing” instead of “enhance” to match “supporting” and “addressing”

Done. Sentence was changed.

Line 108 – incorporate them into what?

Thank you. We have changed the sentence.

Line 119 – “Avoidance” with a capital A to match the other numbers

Thank you. We change it.

Line 217 – Since the use of the ES expert scores comes Before you describe how they were done, I suggest adding something like “as described in Section 2.4” or similar just to be clear

We added the suggestion

Line 233- Just to be clear – the expert panel assigned a value from 0-10 for each of the 25 ES for each USS  (i.e., a total of 25 x 22 scores), which were then averaged by USS to create a mean (?) score for each of P-ES, R-ES, and C-ES?  Did the panel assign a single score, or was each panel member interviewed separately to generate an average score? ETc.   Just a few more details on exactly how the scoring was done would be helpful.

Done. We have improved the text. Please refer to lines 222-228.

Figure 2 – in the two lower graphs I think “Biotop” should be replaced with “Biotope” and “Recreational value”

Done, figure 2 was amended.

Line 315 – this technical recommendation (based on minimizing open space) doesn’t seem to consider other factors when building a road (such as the fact that open space may be the only available property on which to build the road without having to tear down existing infrastructure); A1 seems to run through public buildings and housing; is preserving open space really a priority over preserving existing infrastructure? I think this technical recommendation needs to be more carefully worded here, and this broader issue of tradeoffs between ES and other factors included in the Discussion

Certainly there are other factors. But we followed the recommendations established in the German legislation. To avoid confusion, we have deleted the sentence.

Figure 3 – If the table and graph contain identical information, you really don’t need both; there is a typo in “railroad”; the x-axis on the graph needs a label and units; you could also consistently capitalize the USS labels; also, why are these plotted in reverse order (both alphabetical of USS and the order of A4-A1)

Noted. We have completely amended the figure, including: deleting the table leaving only the bars graph; fixing the typo and capitalizing the labels and adding x-axis label and units

Line 323 – I’m not sure what is meant by “in terms of environmental quality” – I think you are saying urban forest is a higher quality USS than commercial use or arable field, but is this just a general assumption or is it based on the soil/biotope/recreation analysis in 3.1.1. Either clarify what is meant by quality here, or I think wait to discuss quality until the next section?

Noted. We have changed the sentence.

Line 333 – biotope and recreation are mentioned twice in this sentence;

We change the sentence.

Figure 4 – I know soil values were categorized as low to very high, but it is not clear exactly what is meant by high or very high biotope or recreational values – what was the cutoff to be considered high or very high?

This is explained in section 2.3.1. We added the description to avoid confusion:

“Our assessment focused only on the highest occurring biotope values (6-7 high, 8-9 very high), for which we computed the respective hectares.”

Figure 4 – Again, if the table and the figure are identical here, you don’t need both; suggest making labeling consistent with Figure 1 and the main text (A1 instead of Access 1); you also don’t necessarily need the numbers overlaid on the figure bars, as it is fairly easy to interpret the hectare numbers from the graph

Done. We have changed the figure following the suggestions.

Line 346 – it is an oversimplification I think to say “A3 is a worse option” – you are basing this on Open space hectares. If for example, the priority was to preserve Regulating ES or Cultural ES, then A3 might seem like a better option.  It depends on the tradeoffs across R, P, C and open space.

We have deleted the sentence.

Line 347 – “more of open space lost” – by talking about open space, you seem to be mixing your first analysis (open space, biotope, soil, recreation) with your ES analysis; I think it would be better to focus on just the ES results here, and then bring in open space as part of the final compensation synthesis (as described in Line 261);  Otherwise you should clarify in you methods that your standard analysis and ES analysis include each’s respective metrics + open space;

See previous answer.

Figure 5 – Again if table and graph are the same, you don’t need both; you need a y-axis label and units; What are the units on the ES – those are total scores (Ek)?  What is the “average” of – just the ES I assume, but as written it appears to also include Open space;  I wonder if it is necessary to include Open space as part of this figure/table – it was already discussed in Lines 309-315 and is just repeated here;

Thank you for the suggestions. We have changed fig 5 according.

Line 361 – does the 63 cells of the compensation area include the road location? Based on Fig. 2 the road appears to go through the compensation area, but it is difficult to tell

The 63 cells do not include the compensation area, because in such case compensation wouldn’t be possible there.

Figure 6 – You should give ranges on what is meant by very low to very high – i.e., are these based on quintiles or just a uniform division of scores (0-20, 20-40, etc.); These are essentially the Ek scores in each hexagon, right?  You should include in the Figure legend that this is the example for A4.

We have deleted the firs row of maps, according to the request of the next comment.

Figure 6 – It is difficult to visualize how these hexagonal grids overlay on the maps in Figure 2.  Could the road A4 be added? From what I can tell, the road runs through the compensation area?  But the text says the compensation area was selected near the road? I am trying to compare the bottom row to the middle row to see exactly where ecosystem services were lost and where they were compensated for, but it is difficult without visualizing the road location.

Done. The road A4 buffer was added to the map. It is clear that the compensation does not overlap with the road buffer.

Section 3.3. – The way this section is written, it appears you only did the compensation analysis for option A4, as none of the other options are discussed. Could the A1, A2, A3 options be added to Figure 6, so that the reader could see how these options change the landscape?  I realize this would only work if the compensation areas do not overlap;  to save space, I think you could eliminate the top row (the black and white ES scores) as it is somewhat redundant with the middle row;

We have not added the options A1, A2 and 3 for the reason we give previously. However, we have included the A4 for better visualization and we have also deleted the first row of ES maps.

Line 386 – “avoid the temptation to distill into a single number” – although you kind of do this by distilling your 25 ES down to a single ES score, rather than looking at tradeoffs among them individually.  This point is illustrated in line 346 where you declare A4 the winner based on overall ES score (even though A3 wins for both R-ES and C-ES);  If, for example, the community prioritized C-ES more strongly than P-ES, they might prefer A3.  Furthermore, by consolidating your 25 ES metrics to 3 categories, you also potentially lost a lot of information on tradeoffs, assuming all 25 metrics are of equal importance. It is worth a consideration of what was lost or gained by taking this approach (rather than examining metrics individually).

We didn’t summarize into a single number, we rather accounted for the ES bundles using the CICES classification (P-ES, R-ES and C-ES), recognizing that trade-offs occur strongly between these categories (Inostroza 2019).

Inostroza, Luis. 2019. ‘Clustering Spatially Explicit Bundles of Ecosystem Services in A Central European Region Clustering Spatially Explicit Bundles of Ecosystem Services in A Central European Region’. In IOP Conference Series: Materials Science and Engineering, 1–9. https://doi.org/10.1088/1757-899X/471/9/092027.

Furthermore, it would have been largely tediously to compare 25x3 ES. This is actually possible to be done and maybe necessary when biophysical measurement of ES are used alongside public participation, which was not our case. In our case, our aim was to illustrate on a real case study how the ES framework can help improving the existing scheme of compensations used in Germany that follows regulations. Finally, the method would be essentially the same, but with a greater detail in the trade-off analysis, because include marginal value calculation, as suggested by Chan.

Line 387 – typo – capitalize “It”;  Actually the phrasing “it is required” seems too strong;  I think you mean an alternative (better) approach to a single monetary value is a broader consideration multiple ES linked with social considerations; This could be more clearly stated

Thank you for this suggestion. We have changed the text according.

Line 411 – “with contrasting results” – I think there merits further explanation here as to why A4 is the best option in the second analysis, but A1 is in the first;  What additional information is the ES analysis providing that caused the changed in results;

Thank you for your comment. We have added the following paragraph to clarify this point:

- The merit of the ES assessment, as analysed in the introduction (section 1.2) is that brings into the decision making process the hidden or underrepresented values of nature that could be eventually lost. Our analysis also includes the full spectrum of USS and the respective benefits that society receives from ecosystems in such areas.

For example, Recreation in the first analysis and C-ES second analysis are calculated very similarly with the exception of 1 ES (scientific investigation and knowledge);  So I’d expect the results in Figure 4 and Figure 5 to be fairly similar – yet in Fig. 5 A1 has higher recreational value (A2>A1>A4>A3), whereas in Fig. 4 A4 does (A2>A4>A3>A1).  Why the difference?   

The difference is due to the different in size and respective USS in each of the buffers. Our calculations, according to the presented equations, are “area weighted”. This is also clear while looking at the references we have used. Recreation value is higher for A1 (3.2) than A4 (2.8) in figure 4.

Overall the ES analysis has more metrics – so is it possible that the metrics (other than those related to Soil or Biotope) tended to be lower for A4, so as more metrics are included, then A4 starts to look like a better option? In other words, is it really the inclusion of more types of metrics driving the pattern (broad focus on 'provisioning' vs. narrow focus on soil?)?

The real difference lies in the inclusion of ES assessment, that what our differences they show. However, this situation is context specific, as all ES assessment are.

Alternatively, the results of the Standard analysis and ES analysis are based on different measures (hectares of high or very high quality, vs. total ES score) – had the ES analysis been based on hectares of high/very high as well would the results have changed?  Is the inclusion of lower quality ES "hectares" dragging down the A1 score in the 2nd analysis?

ES scores are “area weighted”, this means that we have included the order of magnitude of the impacted space as well, to have a better comparison basis. We could have also used the scores in the case of soil, biotope and recreational value, but as we are basically only prioritising, this is identifying which alternative has higher and lower impact, the inclusion of scores wouldn’t have changed the results. Bottom line the comparison is aimed at showing that the narrow approach of environmental compensation (soil, biotope, recreation) is weaker that the ES because this includes the full spectrum of benefits. That is, at the end of the day, what ES assessments are all about.

To what extent do you think expert opinion to develop the ES scores drove the results (more or less accurately) than the approach to quantifying soil or biotope (which were more data based rather than expert opinion)?  Could the observed differences between the two analysis reflect uncertainty in how the values are measured?

For sure, there are uncertainties in the use of expert assessments. However, as has been analysed by Roche (2020), expert assessment is in high agreement with biophysical measures. We used 11 experts from different disciplines, and calculated the average for each ES. It was dispersion between the individual scores, but we think that the average is able to show the subjacent biophysical trend. We have to remember that we are asking experienced people, which gives certain ground to consider their judgement, as it has been already done in other research.

Kopperoinen L, Itkonen P, Niemelä J (2014) Using expert knowledge in combining green infrastructure and ecosystem services in land use planning: An insight into a new place-based methodology. Landsc Ecol 29:1361–1375. https://doi.org/10.1007/s10980-014-0014-2

Montoya-Tangarife C, Barrera F De, Salazar A, et al (2017) Monitoring the effects of land cover change on the supply of ecosystem services in an urban region : A study of Santiago-Valparaı Chile. PLoS One 12:1–22. https://doi.org/10.1371/journal.pone.0188117

Mukul SA, Sohel MSIMSI, Herbohn J, et al (2017) Integrating ecosystem services supply potential from future land-use scenarios in protected area management: A Bangladesh case study. Ecosyst Serv 26:355–364. https://doi.org/10.1016/j.ecoser.2017.04.001

Etcetera – bottom line I think it is worth more discussion about how/why the approach to the analysis caused the ‘preferred’ solution to change

We have added a new paragraph between lines 405-409. See also previous answer to the same question.

Reviewer 2 Report

The manuscript presented for review concerns the issues related to Ecosystem Services losses in an urban planning context. Ecosytem Services is a very popular and frequently researched issue these years. Increasingly, the attention of researchers is focused on ES related to urbanization and cities. This is a topical topic, as there is an intensive migration of people to cities, and the transformation of suburban areas into places to live for people who work in cities. Often, various types of investments are carried out in cities (e.g. construction of new roads or ring roads), which on the one hand are to relieve the city as a living environment for people, and on the other, they lead to changes in the environment. For this reason, it is so important to combine urban planning with ES. This is what the manuscript is all about. It is, in my opinion, well planned and written. It provides a lot of new and interesting information. My only remark concerns the editorial part of the manuscript - the authors have to adapt it to the requirements of the Land journal (e.g. it concerns the citation of literature in the text of the manuscript). 

Author Response

The authors would like to express their gratefulness for the helpful comments and suggestions made by reviewers to improve the paper. We have taken all remarks carefully into consideration and responded to them accordingly.

Reviewer #2

Comment

Response

The manuscript presented for review concerns the issues related to Ecosystem Services losses in an urban planning context. Ecosytem Services is a very popular and frequently researched issue these years. Increasingly, the attention of researchers is focused on ES related to urbanization and cities. This is a topical topic, as there is an intensive migration of people to cities, and the transformation of suburban areas into places to live for people who work in cities. Often, various types of investments are carried out in cities (e.g. construction of new roads or ring roads), which on the one hand are to relieve the city as a living environment for people, and on the other, they lead to changes in the environment. For this reason, it is so important to combine urban planning with ES. This is what the manuscript is all about. It is, in my opinion, well planned and written. It provides a lot of new and interesting information.

Thank you very much for your positive comments.

My only remark concerns the editorial part of the manuscript - the authors have to adapt it to the requirements of the Land journal (e.g. it concerns the citation of literature in the text of the manuscript).

Thank you. We have changed the reference list following the required format.

Reviewer 3 Report

The paper provides a methodology to assess ecosystem services losses from an urban planning perspective through the example of Bochum, Germany. The topic is current and highly important due to the crucial role of ES in urban areas; however, the manuscript might go through a thorough revision to fill some methodological and structural gap; please see my detailed comments below:

  • the first section might be expanded by adding a more detailed overview of ecosystem services in urban areas. Line 28-37 includes some of them; however, a detailed list is lacking, and it might prepare the ground for further sections;
  • the 1.1. subsection might be completed by providing a more holistic overview of current ES assessments studies. It would be useful to define the scientific added value of the paper in the following subsection;
  • line 104-130 - please consider moving this paragraph to the end of the first section;
  • line 138-149  - please clearly describe why the selected (or elaborated) methodology in unique and useful for further urban development processes;
  • line 174-181 - this paragraph does not include any information about the LULC classification process - please explain how those LULC classes have been identified and measured;
  • Table 1 and Table 2 are rather figures than tables in their present form - please restructure them;
  • the Discussion section is comprehensive; however, the limitations of the methodology is lacking - please add this aspect to the section. Moreover, two further questions might be answered: 1) What new things (new theories, new methods, or new policies) about urban policy or planning can the paper contribute to the existing international literature? The novelty/originality should be clearly justified that the manuscript contains sufficient contributions to the new body of knowledge from the international perspective.  2) How to link the analysis results and main findings in this paper with the previous findings from other countries?
  • Finally, the Conclusions might be expanded by introducing the most important local-specific results of the paper since this aspect lacks the present form of the manuscript.

Author Response

The authors would like to express their gratefulness for the helpful comments and suggestions made by reviewers to improve the paper. We have taken all remarks carefully into consideration and responded to them accordingly.

Reviewer #3

Comment

Response

The paper provides a methodology to assess ecosystem services losses from an urban planning perspective through the example of Bochum, Germany. The topic is current and highly important due to the crucial role of ES in urban areas; however, the manuscript might go through a thorough revision to fill some methodological and structural gap; please see my detailed comments below:

Thank you very much for your positive comment.

the first section might be expanded by adding a more detailed overview of ecosystem services in urban areas. Line 28-37 includes some of them; however, a detailed list is lacking, and it might prepare the ground for further sections;

Thank you for this important remark. We have added the following sentence in the introduction to clarify this point (lines 53-55):

“CICES in the version v5.1 offers a detailed and extensive list of ES that can be applied to identify the relevant services in several geographical settings.

the 1.1. subsection might be completed by providing a more holistic overview of current ES assessments studies. It would be useful to define the scientific added value of the paper in the following subsection;

Readers can refer to the references included. The paper was not meant to provide an extensive review of ES assessment studies. The intended added value of the paper is added (see below).

line 104-130 - please consider moving this paragraph to the end of the first section;

This paragraph is already at the end of the section 1.

line 138-149  - please clearly describe why the selected (or elaborated) methodology in unique and useful for further urban development processes;

Thank you for this remark. We added the following sentence in lines 145-146: “Our method is unique in the sense of articulating ES knowledge with a practical application based on a real planning situation, showing how the ES framework can support decision making.”

line 174-181 - this paragraph does not include any information about the LULC classification process - please explain how those LULC classes have been identified and measured;

We inserted an explanation of USS and the mapping procedure in the manuscript, based on the following quote from [16] “Urban landscapes exhibit intricate patterns of spatial units with different biophysical properties. In order to classify general land use types, Breuste et al. (2013) defined subtypes of residential estates that take different percentages of sealed surfaces into account (cf. Pauleit & Breuste 2011). This refers to earlier approaches (Breuste et al. 2001) to categorize the complexity of the urban fabric by mapping Urban Structural Types. The resulting spatial units provide a basis for assessment and evaluation of UES: They are characterized by comparable eco-environmental properties (features) and are defined according to the actual land use and are further  differentiated by attributes that describe the environmental conditions. Thus, Urban Structural Types enable subdividing hybrid urban landscape mosaics into physiognomically homogeneous units. Mostly, they have an internal characteristic configuration, specific pattern of built-up areas and open space. Each  Structural Type exhibits a characteristic percentage of sealed surface and vegetation structure (text in italics: translated and modified according to Breuste et al. 2001). Urban Structural Types reflect categories of spatial heterogeneity characterized by specific functions and processes.”

Table 1 and Table 2 are rather figures than tables in their present form - please restructure them;

We have changed Table 1 and inserted as a text. For table 2 this is currently not possible as the text is too long. However, we will provide the excel sheets upon for the preparation of the final manuscript layout, as it is the standard procedure.

the Discussion section is comprehensive; however, the limitations of the methodology is lacking - please add this aspect to the section.

Thank you. We added a limitations section.

Moreover, two further questions might be answered: 1) What new things (new theories, new methods, or new policies) about urban policy or planning can the paper contribute to the existing international literature? The novelty/originality should be clearly justified that the manuscript contains sufficient contributions to the new body of knowledge from the international perspective.  2) How to link the analysis results and main findings in this paper with the previous findings from other countries?

We have outlined in the introduction the international arena referring to urban planning and ES. Our original contribution presents a method to articulate in a clear and straightforward manner the existing ES body of knowledge with a real case study of urban development. As we stated in the introduction, considering the forthcoming need to include ES assessments in all policy levels, we believe that our contribution contains sufficient merit to be considered for an international audience interested in the practical application of ES assessments.

In this context, we have also pointed out to the relevant literature in other countries. Specifically referring to ES based expert assessments and mapping approach using the matrix.

Finally, the Conclusions might be expanded by introducing the most important local-specific results of the paper since this aspect lacks the present form of the manuscript.

Thank you for your comment. We added a new sentence in lines 533-534

Reviewer 4 Report

see attachment please

Author Response

The authors would like to express their gratefulness for the helpful comments and suggestions made by reviewers to improve the paper. We have taken all remarks carefully into consideration and responded to them accordingly.

Reviewer #4

Comment

Response

The manuscript raises an important issue of assessing the loss of ecosystem services in the context of urban planning. The aim of this article is to present the methodology and procedure for the implementation of ecosystem services in urban planning based on a case study (part of the city of Bochum).

Thank you for your positive comment.

The authors presented 4 detailed research objectives presented in subchapter 1.2. However, not all goals have been thoroughly researched. The structure of the article with separate chapters is not related to the above-mentioned research objectives. Putting the general research goal and specific goals into different chapters limits the clarity of the article. Detailed objectives appear in section 1.2 concerning the literature review on ES in spatial planning.

Thank you for this comment. We have deleted objective 3 as it corresponds to a former objective which was left out of the scope in the final version of the manuscript.

The work includes multi-faceted research on the problem. The part concerning the literature review, methodology, results and discussion consists of many synthetically presented subsections. In terms of content, the article is prepared correctly, although the methodology of work is not fully explained.

Thank you for this comment. We have improved the method section providing more details on missing aspects.

The authors do not precisely indicate what is innovative in their methodological approach to solving a research problem.

Thank you for this remark. We added a new sentence in lines 145-147 to address this important aspect.

This applies, inter alia, to the criteria for assessing the biotope, soil class and especially recreational values. When assessing a biotope, it is not clear when a space receives a certain score on a scale from 0 to 10. How can a given area receive, for example, 3 points? The classification of soils on a fourpoint scale raises similar uncertainties. Do the evaluation criteria relate to the generally accepted soil valuation, fertility class or agronomic category? The method of assessing the recreational value of the studied area was also too laconic.

This is important. The classification of biotopes is done by experts following an exhaustive a thoughtful procedure largely established in the German planning system. We have used the biotope maps provided by the respective cities. We have only included the few missing values we found within our study area, by identifying similar areas (in terms of vegetation cover, USS, etc.).

Regarding soil, the reviewer is right. Our classes, they do refer to fertility assessed by the combination of classified biophysical parameters. We have included the respective reference which is the soil map for the region.

We added few sentences in the respective sections to improve our description of methods.

The discussion of the research results is partly a continuation of the literature review. There are conclusions that the authors have not studied.

It is a standard procedure to contrast the gaps identified in the introduction with the discussion. We have discussed our approach in the light of the ideas presented in the introduction and we believe that the full narrative of the manuscript is consisting. In the absence of a specific and particular criticism pointing out to lines or paragraphs is difficult to imagine what the reviewer is thinking as misplaced.

The manuscript editorial page has numerous shortcomings.

We did our best is formatting the manuscript. It is however impossible to improve a shortcoming that it is not identified. We don’t know what editorial page means, for instance.

Tables 1 and 2 need to be reworked in accordance with mdpi guidelines.

We have changed Table 1 and inserted as a text. For table 2 this is currently not possible as the text is too long. However, we will provide the excel sheets upon for the preparation of the final manuscript layout, as it is the standard procedure.

Figure 2 may have errors in the description. Is the picture in the lower right corner well described? Does it concern recreation or a biotope?

Thank you for pointing out this mistake. We have fixed the description.

Figures 3, 4, 5 contain tables. Describe tables and figures separately in accordance with mdpi standards. Moreover, figures 4 and 5 are of poor graphic quality.

We have changed all figures and deleted the tables in graphs.

In subsection 1.2, subsection 1.2.1 appears, but there are no further subsections (1.2.2, 1.2.3 etc.).

Yes, that’s correct, because 1.2.1 is a subsection, and the only one, in section 1.2.

The citations of bibliographic sources in the text and the form of the bibliography list are inconsistent with the mdpi guidelines.

We have fixed the citation style.

Round 2

Reviewer 3 Report

The authors replied to all my previous questions and provided adequate and detailed answers; moreover, they modified their manuscript thoroughly. Consequently, the manuscript can be accepted in its present form and structure.